# Effects of Sodium Bicarbonate Ingestion on Recovery in High-Level Judokas

**DOI:** 10.3390/ijerph192013389

**Published:** 2022-10-17

**Authors:** Goran Danković, Nemanja Stanković, Nikola Milošević, Vladimir Živković, Luca Russo, Gian Mario Migliaccio, Alin Larion, Nebojša Trajković, Johnny Padulo

**Affiliations:** 1Faculty of Medical Science, University of Kragujevac, 34000 Kragujevac, Serbia; 2Clinic for Anesthesiology and Intensive Care, University Clinical Center Nis, 18000 Nis, Serbia; 3Faculty of Sport and Physical Education, University of Nis, 18000 Nis, Serbia; 4Department of Human Sciences, Università Telematica Degli Studi IUL, 50122 Florence, Italy; 5Sport Science Academy, 09131 Cagliari, Italy; 6Faculty of Physical Education, Ovidius University of Constanta, 900029 Constanta, Romania; 7Department of Biomedical Sciences for Health, University of Milan, 20100 Milan, Italy

**Keywords:** supplementation, placebo, performance, judokas

## Abstract

Background: The aim of this study was to examine the sodium bicarbonate (NaHCO3) effect on recovery in high-level judokas. Methods: The sample of participants consisted of 10 male judokas (Age = 20 ± 2.1 years) who are judo masters (black belt holders) with a minimum of 10 years of training and competition experience. The study was designed as a double-blinded crossover design with the order of treatments being randomly assigned. The washout period was 72 h. All subjects received a dose of sodium bicarbonate (0.3 g/kg body weight) or a placebo 120 min before the fatigue caused by the special judo fitness test (SJFT). Lactate concentration (LC), countermovement jump (CMJ), hand grip strength and degree of perceived fatigue on Borg’s scale (RPE) were tested two times before SJFT and four times after SJFT. Results: There was no interaction between groups and type of recovery at any time during the two types of recovery for RPE, grip strength, VJ and lactate concentration (*p* > 0.05). However, there was a main effect of time for dominant grip strength (F_(1,8)_= 3.3; *p* = 0.01; η^2^ = 0.25, (small)), non-dominant grip strength (F_(1,8)_ = 3.2; *p* = 0.01; η^2^ = 0.24, (small)), CMJ (F_(1,8)_ = 8.8; *p* = 0.01; η^2^ = 0.47, (small)), and LC (F_(1,8)_ = 124.2; *p* = 0.001; η^2^ = 0.92, (moderate)). Conclusions: The results of the present study show no significant difference between the NaHCO3 and placebo groups in RPE, handgrip strength, CMJ, and lactate concentration.

## 1. Introduction

Judo is one of the most popular Olympic combat sports, requiring good tactics, fine technique, and outstanding physical fitness [1,2]. As a grappling combat sport in which opponents attempt to throw or immobilize each other to win the match, a high level of technical and tactical development is needed [3]. Hence, physical conditioning is crucial in judokas, regardless of weight category [4]. As such, it is in the sport’s nature to require activity at submaximal and maximal intensity, with very short recovery time between fights [5,6]. The most significant technique is grip dispute, to disrupt the balance and toss the opponent [7]. That is why the maximal physical strength, endurance, and anaerobic power are required to win a judo-match [8,9,10]. These fitness parameters should be developed by a scientifically supported periodic training program as well [11,12]. In this regard, the energy requirements of a competitive match must be investigated to establish guidelines for developing athletes and increasing their training and recuperation [13].

Recognizing that an athlete might compete in more than five matches on the same day with a 10-min break between two consecutive matches, proper recuperation is critical to competitive performance [14]. Likewise, the judo-match consists of high-intensity intermittent movements, with periods of exertion nearing 30 s and breaks of around 15–30 s [15]. Therefore, to the best of our knowledge, the mentioned break periods may be short for effective ATP activation in order to accelerate the recovery. In accordance with the previously mentioned facts, the supplements usage may postpone the onset of exhaustion, allowing a judoka to train at a higher intensity [16]. Nevertheless, there is also a data scarcity on judokas energy balance, where additional attention must be paid to providing enough energy for basal metabolism, physical activity and most importantly, body and energy recuperation, as well [17]. Therefore, the majority of studies have investigated the effects of different ways of recovery in sport, especially in judo.

To enhance the recovery process, judokas have showed affinity to various types of product usage to speed up the recovery process. Chocolate milk [18], beta-alanine supplementation [19,20], caffeine-containing energy drinks [21], caffeine [22,23], creatine [24], as well as sodium bicarbonate [25,26] seem to be the most effective. From all mentioned, sodium bicarbonate (NaHCO3) has a direct effect on lactic acid, which delays fatigue and tiredness, shortens the recovery time, as well as improving overall performance [27,28,29,30] and increasing blood lactate concentration [31]. Since the carbon dioxide (CO_2_) and hydrogen (H+) accumulate in the blood and muscles during high-intensity exercise, the bicarbonate system rids the H+ and CO_2_ of the body through the renal and respiratory systems by converting them to bicarbonate [32].

However, the usage of NaHCO3 has showed heterogenic results on judokas’ recovery. Šančić et al. [28] results showed that lactic acid levels were significantly lower when judokas have used active recovery versus NaHCO3 intake. On the other hand, Artioli et al. [31,33] have shown that the usage of NaHCO3 has improved judo-related performance and blood lactate concentration, but without visible effects on perceived exertion and recovery. Based on the above-mentioned facts, in-depth investigation research is needed on this topic. Therefore, we have aimed to examine the sodium bicarbonate (NaHCO3) effect on recovery in high-level judokas. We have hypothesized that the supplement usage group will show improved judo performance and enhanced recovery.

## 2. Materials and Methods

### 2.1. Participants

The sample of participants consisted of 10 male judokas (Age 20 ± 2.1 years; Body height 180.18 ± 8.11 cm; Body mass 85.24 ± 23.17 kg; BMI 25.2 ± 3.4 kg/m^2^) who are judo masters with a minimum of 10 years of training and competition experience. Judokas competed in an international level competition and had a minimum black belt and included in the first ranking national level list. All research procedures were carried out in accordance with the International Ethical Guidelines. In addition, the approval (04-1847/2; date of approval 26 November 2020) of the Ethics Committee from the Faculty of Sport and Physical Education, University of Nis, was secured, as well as the written consent of the participants for the inclusion in the studies. The inclusion criteria were male, age (18–25 years), judo master (black belt holders) with minimum 10 years of training and competition experience, absence of disease and injuries and absence of using any supplementation 2 months before the start of this study.

### 2.2. Procedures

The study was designed as a double-blinded crossover design with the order of treatments being randomly assigned. The washout period was 72 h. All subjects received a dose of sodium bicarbonate (0.3 g/kg body weight) [31] or a placebo (Ringer’s solution) 120 min before the fatigue caused by the special judo fitness test, which was validated in high-level judokas [8]. The SJFT was used because of the similar time structure to that of a judo match, and because it can simulate the demands of specific tasks in matches and combat training. Additionally, the participants were instructed to arrive at the tests with instructions to eat similarly in the evening, as well as to avoid intense exercises 24 h prior the testing. Moreover, they were instructed not to ingest of any kind of food in the two hours which preceded the Na-HCO3 intake.

#### 2.2.1. Special Judo Fitness Test

The special judo fitness test was performed in the following order: Two subjects (ukea) of the same weight category and similar height were positioned at 6 m from each other, while the tested subject (tori) was in the middle between them. At the timekeeper’s signal, the tori runs to one uke, make a throw and then make the same throw on the other uke. The test consists of three parts (15, 30 and 30 s) with 10 s recovery between them. The total number of throws during each of the three periods was recorded: immediately after the end of the third part, the heart rate was measured with a heart rate monitor (after exercise) as well as after 60 s of rest (after recovery). The index was calculated by summing the results of the heart rate after the test and the heart rate after 60 s of recovery, which is related to the total number of throws (n).
Index = HR after the test + HR 1 min after the test/n

Measurements of anthropometric parameters (body height, body weight) were measured using anthropometer and the bioelectrical impedance method (InBody 770, InBody USA, Cerritos, CA, USA) before testing.

Procedures for the assessment of blood lactate analysis was performed according to the standard protocol with the help of the Lactate Scout Analyzer (Accusport TM, Lactate Pro TM, YSI 1500 Sport, Boehringer Mannheim, Germany) device. Lactate analysis was performed six times during the study (1—prior supplementation, 2—after supplementation, 3—immediate after first SJFT, 4—after first SJFT (5 min.), 5—after first SJFT (30 min.), 6—immediate after second SJFT). Heart rate was monitored using a heart rate monitor (Polar Team System Pro, Kempele, Finland).

Assessment of fitness parameters was performed in the same timeline as for the blood lactate. Countermovement jump (CMJ), and Hand grip strength were tested two times before SJFT and four times after SFJT. Second SJFT was conducted 30 min after the first, simulating the combats timeline during competitions. During every time point, all subjects marked the degree of perceived fatigue (RPE) on Borg’s scale (using the 0–10).

#### 2.2.2. Countermovement Jump (CMJ)

“Optojump” (MicroGate, Bolzano, Italy) was used to measure the explosive power of the lower extremities. Countermovement jump (CMJ) was performed as follows:

The initial position of the examinee was standing upright, with feet hip-width apart, hands with palms resting on the examinee’s hips. The participant’s task is to quickly lower himself from the initial position to a squatting position at a 90° angle between the lower and upper leg. Without taking a break, the participant jumps as high as possible and lands on the ground with both feet at the same time. The result of the test is the height of the jump expressed in centimeters, measured with the help of the Optojump device. The best out of three jumps was used for further analysis. Validity and reliability were presented elsewhere [34].

#### 2.2.3. Handgrip Strength

The force realized at the maximum handgrip was measured using an electronic manual dynamometer (Uno Lux, Belgrade, Serbia). The hand dynamometry was performed by participant holding the dynamometer in a slightly bent hand at the elbow joint and trying to achieve the best possible result with a maximally strong grip. Each participant performed the test twice with the stronger and weaker hand, and the better result was recorded as relevant for statistical processing. Results were presented in Newtons (N) with a measurement accuracy of 0.01 N.

### 2.3. Study Power and Sample Size

The calculation of the total sample was based on the results of a previously published study of a similar design [3]. Calculation of the required sample size was performed using the software package G*Power v.3.1.9.2 (Kiel, Germany). The statistical power of the research was set at 80% (β ≤ 0.2), and the confidence level at 95% (r < 0.05). A small difference in the dependent variables of physiological and motor abilities between the two different test conditions (sodium bicarbonate supplementation versus placebo) was expected, which can be represented by a small to moderate effect size represented by a value of 0.3. Moreover, as an input parameter, the expected large correlation coefficient between two different tests was entered, represented by the value 0.3. Under the mentioned conditions, the required size of the final sample was estimated at *n* = 10.

### 2.4. Statistical Analysis

To test the normality of the data, we have applied the Kolmogorov–Smirnov test. Additionally, the data were tested for homogeneity using Levene’s test. After that, a two-way analysis of variance (ANOVA) with repeated measurements was used with supplementation and time point as factors (supplementation × round). When necessary, a post hoc test was used to identify possible differences between conditions and time. For ANOVA results, effect sizes were calculated using eta squared (η^2^), classified according to Cohen. The level of statistical significance was set at *p* < 0.05. Statistical data processing was carried out in the statistical package IBM SPSS 24 (version 24.0; SPSS, Inc., Chicago, IL, USA).

## 3. Results

Table 1 presents RPE, grip strength, CMJ and lactate concentration in each group during different time points. The results were also presented graphically (Figure 1). For RPE, there was a main effect of time (F_(1,8)_ = 43.3; *p* = 0.001; η^2^ = 0.82, (moderate)), with values after the second time point significantly higher in relation to before in both conditions (Table 1). There were no effects of group (*p* > 0.05) or supplementation × round interaction (*p* > 0.05). There were no interactions between groups and type of recovery at any time during the two types of recovery for RPE, grip strength, CMJ and lactate concentration (*p* > 0.05). However, there was a main effect of time for dominant grip strength (F_(1,8)_ = 3.3; *p* = 0.01; η^2^ = 0.25, (small)), non-dominant grip strength (F_(1,8)_ = 3.2; *p* = 0.01; η^2^ = 0.24, (small)), CMJ (F_(1,8)_ = 8.8; *p* = 0.01; η^2^ = 0.47, (small)), and LC (F_(1,8)_ = 124.2; *p* = 0.001; η^2^ = 0.92, (moderate)).

## 4. Discussion

The present study was designed to examine the effect of sodium bicarbonate (NaHCO3) on lactate concentration and performance recovery after a special judo fitness test. The main findings were that the judokas who underwent sodium bicarbonate showed similar results in RPE, handgrip strength, CMJ, and lactate concentration compared to the placebo group, without significant differences in each time point of the recovery.

The RPE method has been demonstrated to be an effective and practical tool for researchers and athletes to track training intensities [35]. In the current study, an SJFT was administered, and the subject’s RPE was used to monitor their training load. There were no significant differences for RPE between the experimental and control groups, which confirms that similar training loads were administered. Since our results have presented lower values of RPE in the sodium bicarbonate group immediately after the first SJFT, 5 min. after SJFT and after the second SJFT, several studies could be related to our results [6,31]. There are similarities in the results of our study compared to Zabala et al. [36], who have also examined the RPE right after their Wingate test; however, the sodium bicarbonate had no effect. Although there was sport disparity (judo vs. BMX cycling), the same disparity could be related to the metabolic demands of the chosen exercise. The possible mechanism behind the link between acid–base balance and RPE during exercise may entail the negative effects of intracellular H+ buildup on muscular force-generating capacity when the muscle gets the fatigue stage [37]. Hence, future studies are needed as well.

High-performance endurance athletes frequently have their blood lactate response to exercise measured as part of their physiological evaluation [38]. Likewise, the current study has presented that the usage of sodium bicarbonate has increased plasma lactate in the experimental group. Increased post-exercise lactate concentration has been also reported in other similar investigations as well [26,31,39,40]. Our results could be compared with a result of Artioli et al. [31], where judokas had performance consisted of attempting as many throws as possible within three discrete time periods (which can be related to our special judo fitness test). Their results suggested that the sodium bicarbonate group performed significantly more throws (5.1%; *p* < 0.01), as well as the higher relative mean and peak power on the Wingate test (*p* < 0.05), regarding the placebo group. To explain mechanism for the given results, the possible explanation could be in the fact that increased H+ efflux relates to a bigger bout of lactate transport to the extracellular surroundings [41]. Regarding the possible mechanism, this perfectly explains the increased glycolytic activity as well, which is later related to increased performance [25]. However, to completely understand the obtained results, the future ones are needed as well, because different exercise protocols, along with different doses of sodium bicarbonate have reached different understandings, whereas blood lactate concentration have presented non-significant changes as well [42,43].

Handgrip strength is a key factor in judo performance, allowing the use of throwing techniques and influencing the opponent’s movements [44]; additionally, better explosive power in lower limbs can increase the number of throws made by a judoka [19]. Our results show no significant difference in physical performance (CMJ and handgrip) while ingesting sodium bicarbonate compared to placebo. Similar to our results, in the study of Fellipe et al. [25] there were no improvements in the number of throws in SJFT compared to the placebo group (*p* = 0.22, ES = 0.48) while using NaHCO3 supplement. On the other hand, one study showed that sodium bicarbonate increased the number of throws in the SJFT [31]. Moreover, Bishop et al. [39] showed that NaHCO3 improved performance in repeated sprint ability. Present studies show heterogeneity in the results of sodium bicarbonate improving physical performance in judo athletes. To the best of the authors’ knowledge, little research was based on sodium bicarbonate improving vertical jump performance and handgrip strength. Moreover, according to Danković [45], 0.3 g/kg was the most frequent dose of NaHCO3 acute intake in judokas. Therefore, we are not able to compare and discuss the impact of different dosses on performance in judo. Accordingly, there is a need for different ingestion protocols to obtain some more insights into recovery in judo and other combat sports.

Our study has some limitations. Years of judo practice indicates similarities in judokas, practicing this activity for the same numbers of years. Secondly, the number of subjects was only 10, suggesting lower study power. Additionally, it was challenging to compare research due to a lack of comparable variables, such as the number of throws during SJFT.

## 5. Conclusions

In conclusion, the results of the present study show no significant difference between the NaHCO3 and placebo groups in RPE, handgrip strength, CMJ, and lactate concentration. It has been reported that sodium bicarbonate could improve performance in the laboratory setting. However, this does not necessarily apply to performance in the field, as indicated in the current study. Therefore, further studies are warranted to investigate the effect of sodium bicarbonate on recovery during a competition day or following judo-specific activities.

## Figures and Tables

**Figure 1 ijerph-19-13389-f001:**
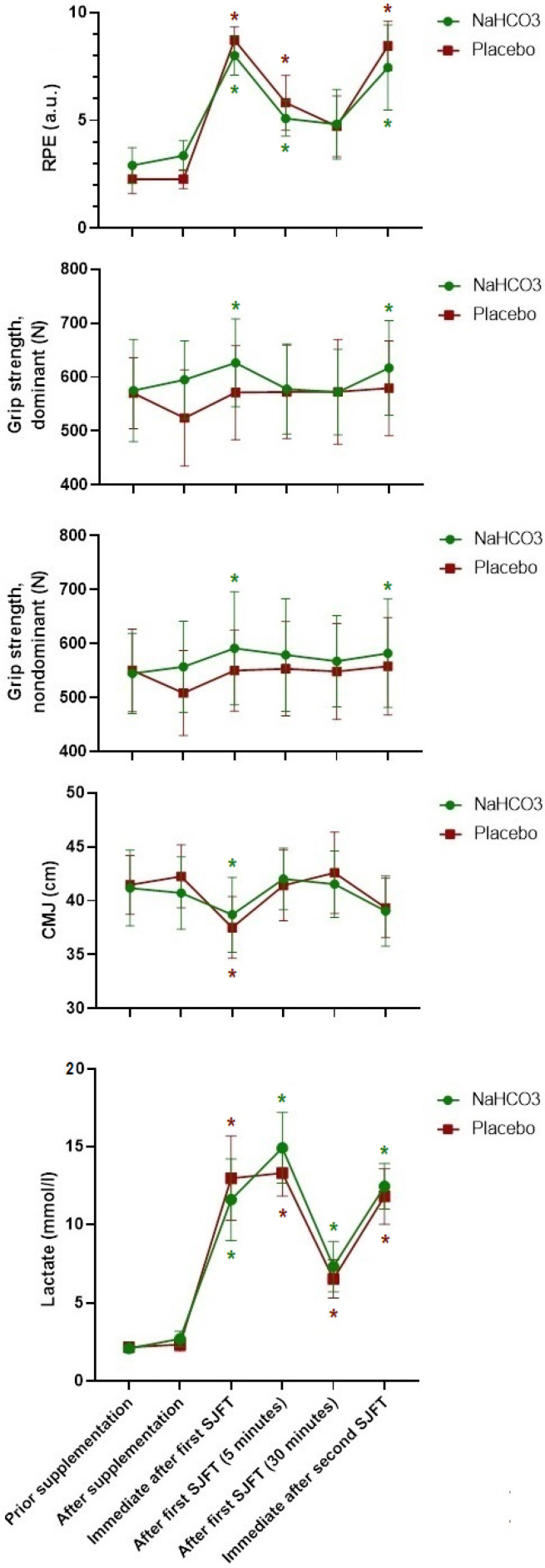
Performance and lactate responses before and after an SJFT. Significance differences with *p* < 0.05 (*) compared to “After supplementation”.

**Table 1 ijerph-19-13389-t001:** Descriptive statistics of the variables studied.

Variables	Group	Time Points					
		1st	2nd	3rd	4th	5th	6th
RPE (a.u.)	NaHCO3	2.9 ± 1.2	3.4 ± 1.0	8.0 ± 1.3 *	5.1 ± 1.2 *	4.8 ± 2.4	7.5 ± 2.9 *
	Placebo	2.3 ± 1.0	2.3 ± 0.6	8.7 ± 0.9 *	5.8 ± 1.9 *	4.7 ± 2.1	8.5 ± 1.7 *
Handgrip strength, dominant (N)	NaHCO3	575.1 ± 140.8	595.1 ± 107.4	626.4 ± 121.4	577.9 ± 124.5	572.3 ± 118.3	617.2 ± 130.6
	Placebo	570.2 ± 98.3	524.2 ± 132.9	571.4 ± 129.7	572.6 ± 129.1	572.8 ± 144.4	579.4 ± 131.4
Handgrip strength, nondominant (N)	NaHCO3	544.8 ± 110.5	557.0 ± 125.9	591.6 ± 155.9	579.0 ± 155.6	567.4 ± 126.0	582.2 ± 149.6
	Placebo	550.5 ± 114.1	508.6 ± 117.4	550.2 ± 111.4	553.5 ± 130.0	548.5 ± 132.4	558.2 ± 134.4
CMJ (cm)	NaHCO3	41.2 ± 5.3	40.7 ± 5.0	38.7 ± 5.2	42.0 ± 4.3	41.5 ± 4.6	39.0 ± 4.8
	Placebo	41.5 ± 4.1	42.3 ± 4.4	37.5 ± 4.3	41.4 ± 4.9	42.6 ± 5.6	39.3 ± 4.1
Lactate (mmol/L)	NaHCO3	2.1 ± 0.3	2.7 ± 0.7	11.6 ± 3.9 *	14.9 ± 3.4 *	7.3 ± 3.4 *	12.4 ± 2.2 *
	Placebo	2.2 ± 0.2	2.3 ± 0.6	12.9 ± 3.9 *	13.3 ± 2.2 *	6.5 ± 1.8 *	11.8 ± 2.6 *

Values are Mean ± SD. Legend: SJFT—senior judo fitness test, 1—prior supplementation, 2—after supplementation, 3—immediate after first SJFT, 4—after first SJFT (5 min.), 5—after first SJFT (30 min.), 6—immediate after second SJFT; CMJ—Countermovement jump; RPE—rate of perceived exertion; *—significant difference compared to the 1st.

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
