# Peer review of "Effects of Sodium Bicarbonate Ingestion on Recovery in High-Level Judokas"

_ijerph, 2022, doi:10.3390/ijerph192013389_

Round 1

Reviewer 1 Report

The manuscript covers a modest topic, which may be useful for judokas because of its applied aspect.  Essentially, sodium bicarbonate has been shown not to be effective for recovery in judokas. 

The research methods, introduction, and discussion are fine. The paper, as a whole, is modest in its effect on the literature, but some may value it because of its negative findings. Publishing negative results is important.

The paper has numerous grammatical and formatting errors that need to be addressed because I suspect the authors are not native speakers of English. I would recommend that the paper be copy edited by a professional editor. Here are some examples of issues that need to be addressed:

Line 2. High-level Judo Athletes instead of High Level...

Lines 2 and 17. Should the authors use judo athletes or judokas? It is not a big issue, but I think the paper should be consistent throughout or define a judoka early as an expert in judo and then go from there.

Line 19. Double-blinded should be hyphenated.

Line 21. All subjects received, not all subjects have received.  The article needs to be written in the past tense for things that occurred in the past.

Line 34. Sport should be sports, as Judo is but one of the most popular Olympic combat sports.  TKD, boxing, and wrestling are others.

Line 42. The comma after required in not necessary, and "This" fitness parameters is incorrect.  These fitness parameters would be correct because more than one parameter is being referenced.

Line 45. "Growing" seems to be the wrong term.  Perhaps developing would be better.

Line 52. A judoka instead of an judoka.

Line 56. Drop "there is a great" and the following "that" and state, "Therefore, a majority of studies have investigated..."

Line 61. Is possess the best word choice here?

Line 68. "But also" is redundant.  Use "However" or "But" or "Also" or just rewrite the sentence to be more clear.

Line 71. Is shown a better word to use than showed?  Is NaHCO3 singular?  Is so, the verb has should be used instead of have.

Line 80. Perhaps the authors could included the calculated BMI (Body Mass Index) as well?

Line 82. The word "in" is repeated.

Line 91. All subject received...  The "have" is not necessary.

Lines 96-98. Check the verb tenses.

Line 117. Assessment is singular and requires a singular verb such as was instead of were. The prepositional phrase "of fitness parameters is throwing off the authors.  This is a tricky one grammatically.

These are just some examples that a quality editor can fix.  Many others exist, but I am not going to edit the entire document as a reviewer. 

Author Response

September 26, 2022

IJERPH

Ref: Submission ID 1931297

Dear Editor,

Please find attached the revised version of our manuscript, entitled “"Effects of Sodium Bicarbonate Ingestion on Recovery in High –  Level Judo Athletes”

We thank the reviewers for their careful evaluation and helpful comments to our manuscript. We have carefully taken their comments into consideration in preparing our revision, which has resulted in a paper that is clearer, broader and more compelling.

Some changes were made to this paper in order to improve it. Please find below our point-by-point responses to each of the comments of the reviewers. All changes are marked in the text.

We hope that the revisions in the manuscript and our accompanying responses will be sufficient to make our manuscript suitable for publication in IJERPH.

Sincerely,

Prof Dr Johnny Padulo

Reviewer 2 Report

The article is interesting. It is methodologically well laid out. However, there are some major concerns.

Minors

Line 50- One should be careful with this type of statement. It is better to say, "to the best of our knowledge...". Prior creatine intake may help ATP-PRC resynthesis.

Major concerns

1.     I am concerned about the number of the sample for the external validity of this study to be representative. Indicate at what level the judokas compete.

2.     Procedures: Why is that amount of sodium bicarbonate used. It would be important to reference some articles that have previously used that amount.

3.     In the specific judo test, has this test been used previously in the scientific literature? What scientific validity does it have? If so, it is necessary to cite the validation article.

4.     In the description of the test, what do you mean by the term "throwing", throwing a object?

5.     Please, in section 5.2, put the text justified.

6.     It is necessary to cite the reference article of the CMJ and SJ tests.

7.     Table 1. Please replace "Grip" with "Handgrip". Indicate the meaning of "*" in the table. The effect size, as well as the p-value, should appear in the table. SJ data are not shown. Also, in the paragraph prior to Table 1 it is referred to as VJ, vertical jump?

8.     When comparing with other research in the discussion section, it is important to put the doses administered in the other studies.

9.     Could a higher dose have contributed to a greater response? This should be addressed in the discussion with scientific support.

Author Response

(The authors gave the same response as above.)

Reviewer 3 Report

Thank you for your hard work. This study is very practical and meaningful.

There are some shortage of the study design.

Major:

The washout period was 2–7 days, which was too casual to apply a scientific study.

How to control the fatigue of each subject to the same degree just 120 minutes after the administration of the supplement was not mentioned, and it is difficult to accurately control .

The fatigue standard was not clear. How to control the same fatigue degree in two stage(intake or not intake) is not clear.

The time of lactate collection was inappropriate. Lactate clearance is more effective on assessment of fatigue.

How many CMJ tests have you done for each participant? Which data did you choose for the statistical analysis? Meanwhile, previous study reported that the average CMJ height was more sensitive than highest CMJ height in monitoring neuromuscular status. (Claudino JG et al. DOI: 10.1016/j.jsams.2016.08.011)

The basis of the dose of the NaHCO3 supplement was not sufficient.

Minor:

The interval time between SJFT 1 and SJFT 2 is not clear.

The basis of the SJFT was not sufficient.

Six times of tests were conducted in the current study, but only five are described in the abstract.

Author Response

(The authors gave the same response as above.)

Round 2

Reviewer 2 Report

Accept in present form. Congratulation.

Author Response

Thanks to endorse our article

Reviewer 3 Report

Thank you for your hard work again.

"The washout period was 2–7 days." was described in the old version, now it was described as "The washout period was 72h", which was too casual to write a scientific paper.

Judo masters (black belt holders) was not high-level athletes. There are 10 degrees in judo, black belt was the fifth degree, which was an advanced beginner.

"How to control the fatigue of each subject to the same degree just 120 minutes after the administration of the supplement was not mentioned, and it is difficult to accurately control.", which was not appropriate controlled.

The RPEs(8.7±0.9, which was the biggest figure in this paper,) was too low (RPE is 0-20), so the athletes were not fatigue.

The basis of the dose of the NaHCO3 supplement was not sufficient.

The time of lactate collection was inappropriate. Lactate clearance is more effective on assessment of fatigue.

* was added in the table 1, but the description of the * was insufficient. 

The information of the Figure 1 was not sufficient, there was no significant symbol.

Author Response

We thank the reviewer for the careful evaluation and helpful comments to our manuscript. We have carefully taken the comments into consideration in preparing our revision, which has resulted in a paper that is clearer, broader and more compelling.

yes I just read: Thank you for your hard work again.

R: thanks for your support to improve the main document

"The washout period was 2–7 days." was described in the old version, now it was described as "The washout period was 72h", which was too casual to write a scientific paper.

R: According with your previously suggestion to avoid any confusion we calculate the hours corresponding on 3 days (72h)

Judo masters (black belt holders) was not high-level athletes. There are 10 degrees in judo, black belt was the fifth degree, which was an advanced beginner.

R: We included more details to better clarify the judoka’s level

"How to control the fatigue of each subject to the same degree just 120 minutes after the administration of the supplement was not mentioned, and it is difficult to accurately control.", which was not appropriate controlled.

R: as showed in Figure 1 (RPE and Lactate was lower and similar for both groups) in the first two time points (Prior and after supplementation)

The RPEs(8.7±0.9, which was the biggest figure in this paper,) was too low (RPE is 0-20), so the athletes were not fatigue.

R: thanks for this suggestion, it was a mistake because the RPE scale correct is 0-10 / therefore RPE ~8 represent a higher Rating Perception Effort

The basis of the dose of the NaHCO3 supplement was not sufficient.

R: We calculate the dose (0.3 g/kg body weight) according to the Artioli GG, Gualano B, Coelho DF, Benatti FB, Gailey AW, Lancha AH. Does sodium-bicarbonate ingestion improve simulated judo performance?. International Journal of Sport Nutrition and Exercise Metabolism. 2007 Apr 1;17(2):206-17.

The time of lactate collection was inappropriate. Lactate clearance is more effective on assessment of fatigue.

R: According with the recent literature we included RPE and Lactate

* was added in the table 1, but the description of the * was insufficient.

R: We included in Table 1 all data for both conditions and for all variables

The information of the Figure 1 was not sufficient, there was no significant symbol.

R: the figure was updated (y axes for Lactate and we added the significance values for both groups)